# Preserving Ambulation in a Gene Therapy-Treated Girl Affected by Metachromatic Leukodystrophy: A Case Report

**DOI:** 10.3390/jpm13040637

**Published:** 2023-04-06

**Authors:** Silvia Faccioli, Silvia Sassi, Daniela Pandarese, Corrado Borghi, Valentina Montemaggiori, Marina Sarzana, Stefano Scarparo, Carla Butera, Valeria Calbi, Alessandro Aiuti, Francesca Fumagalli

**Affiliations:** 1Children Rehabilitation Unit of S. M. Nuova Hospital, Azienda Unità Sanitaria Locale IRCCS di Reggio Emilia, 42121 Reggio Emilia, Italy; 2PhD Program in Clinical and Experimental Medicine, Department of Biomedical, Metabolic and Neural Sciences, University of Modena and Reggio Emilia, 41100 Modena, Italy; 3Orthopaedic Unit of S. M. Nuova Hospital, Azienda Unità Sanitaria Locale IRCCS di Reggio Emilia, 42121 Reggio Emilia, Italy; 4Pediatric Immunohematology Unit, IRCCS San Raffaele Scientific Institute, 20019 Milan, Italy; 5Units of Neurology and Neurophysiology, IRCCS San Raffaele Scientific Institute, 20019 Milan, Italy; 6San Raffaele Telethon Institute for Gene Therapy (SR-TIGET), IRCCS San Raffaele Scientific Institute, 20019 Milan, Italy; 7Vita-Salute San Raffaele University, 20019 Milan, Italy

**Keywords:** rehabilitation, orthopedics, botulinum toxins, muscle spasticity, assistive devices, gait analysis, disabled children, nerve conduction studies

## Abstract

(1) Background: Atidarsagene autotemcel is a hematopoietic stem and progenitor cell gene therapy (HSPC-GT) approved to treat early-onset metachromatic leukodystrophy (MLD). The purpose of this case report is to describe the long-term management of residual gait impairment of a child with late infantile MLD treated with HSPC-GT. (2) Methods: Assessment included Gross Motor Function Measure-88, nerve conduction study, body mass index (BMI), Modified Tardieu Scale, passive range of motion, modified Medical Research Council scale, and gait analysis. Interventions included orthoses, a walker, orthopedic surgery, physiotherapy, and botulinum. (3) Results: Orthoses and a walker were fundamental to maintaining ambulation. Orthopedic surgery positively influenced gait by reducing equinovarus. Nonetheless, unilateral recurrence of varo-supination was observed, attributable to spasticity and muscle imbalance. Botulinum improved foot alignment but induced transient overall weakness. A significant increase in BMI occurred. Finally, a shift to bilateral valgopronation was observed, more easily managed with orthoses. (4) Conclusions: HSPC-GT preserved survival and locomotor abilities. Rehabilitation was then considered fundamental as a complementary treatment. Muscle imbalance and increased BMI contributed to gait deterioration in the growing phase. Caution is recommended when considering botulinum in similar subjects, as the risk of inducing overall weakness can outweigh the benefits of spasticity reduction.

## 1. Introduction

Metachromatic leukodystrophy (MLD) is an inherited lysosomal storage disease caused by arylsulfatase A (ARSA) deficiency that results in progressive demyelination of the central and peripheral nervous system. Children affected by the early-onset variants (Late Infantile—LI and Early Juvenile—EJ) show progressive motor and cognitive impairment, leading to severe disability and fatal outcomes a few years after symptoms manifest [1,2,3]. The natural course of the LI variant is characterized by a very rapid loss of motor and cognitive function due to concurrent central and peripheral demyelination and neurodegeneration, leading affected children to be bedridden by the age of 3 years [2,3,4,5,6,7,8]. In recent years, ex vivo lentiviral hematopoietic stem cell gene therapy (HSPC-GT) has been proven to provide clinical benefits in MLD. After promising results were obtained in MLD mouse models [9,10], a non-randomized, open-label, single-arm phase I/II clinical trial of HSPC-GT for the treatment of MLD was started in Milan, Italy, in 2010. HSPC-GT consists of autologous CD34+ cells transduced ex vivo with a lentiviral vector encoding for the human ARSA gene, administered intravenously following myeloablative busulfan conditioning. The preliminary results on three presymptomatic LI children showed stable engraftment of the transduced hematopoietic stem cells (HSC) in the bone marrow and peripheral blood [11], with reconstitution of ARSA activity in all hematopoietic lineages and also in the cerebrospinal fluid (CSF), providing indirect evidence that HSC-derived cells had migrated to the central nervous system (CNS) and had produced the enzyme locally. Further results on the first 9 treated patients (6 LI, 2 EJ, and 1 with an intermediate phenotype) [12], with a median follow-up of 36 months, underlined that the extent of benefit was influenced by the time interval between GT and the expected time of disease onset. In fact, better results were obtained in children treated when either presymptomatic or very early after onset, with maintaining motor and cognitive functions and preventing or delaying CNS demyelination.

Integrated analyses were recently published on the first 29 presymptomatic or early symptomatic, early-onset subjects treated with HSPC-GT with a median follow-up of 3 years (range: 0.64–7.51) [13]. Long-term follow-up of this larger cohort of patients confirmed sustained multilineage engraftment of genetically modified HSC and persistent increase in ARSA activity in peripheral blood and CSF. Most importantly, it has been demonstrated that 2 years after treatment, gross motor function measures of treated patients were significantly higher compared to age-matched and disease subtype-matched natural history (NHx) patients for both subjects with LI and with EJ MLD. Most treated patients progressively acquired motor skills following normal motor development, experiencing stabilization of motor performance (maintaining the ability to walk over the long term) or a delay in the rate of the progression of motor dysfunction. Moreover, HSPC-GT was able to preserve normal cognitive development and prevent or delay central and peripheral demyelination and brain atrophy. Of note, treatment benefits were particularly apparent in patients treated before symptom onset.

Based on the outcomes described in this ad hoc interim analysis, HSPC-GT (Atidarsagene autotemcel) was approved in Europe in December 2020 https://www.ema.europa.eu/en/medicines/human/EPAR/libmeldy (accessed on 1 September 2022), and the treatment is currently available in selected qualified treatment centers in Europe and the United Kingdom.

Despite clear evidence that HSPC-GT constitutes a disease-modifying treatment, the characterization and management of the residual disease burden have not yet been described in the literature.

We report the case of a girl affected with MLD who was admitted for rehabilitation in June 2017 at the age of 6 years because of a worsening gait. Given her family history, the patient was diagnosed with presymptomatic late infantile MLD at the age of 1 year based on biochemical and genetic evidence. Immediately after diagnosis, she was given the opportunity to be enrolled in a therapeutic clinical trial (ClinicalTrials.gov Identifier: NCT01560182) underway at the Ospedale San Raffaele in Milan, approved by the Ospedale San Raffaele Ethics Committee and by the Agenzia Italiana del Farmaco. At the age of 15 months, the patient underwent treatment with atidarsagene autotemcel [12]. After infusion and complete hematological recovery, biochemical, clinical, neurophysiological, and neuroradiological follow-up showed a satisfactory stable situation up to 2017, at the age of 6 years (5 years post gene therapy), when, despite unchanged biochemical and instrumental data, the patient began to exhibit increasing gait difficulties. The innovative contribution of this case report is that it deepens the understanding of the causes and the management of residual disease burden related to gait impairment and describes the results of subsequent therapeutic approaches by means of computed gait analysis.

## 2. Detailed Case Description

The timetable of interventions and assessments is represented in Table 1.

Clinical and X-ray images of the patient are presented in Figure 1. Figure 1A shows initial, still flexible varo-supination and excessive hyperextension of the knee at first access to the rehabilitation unit (age 75 months). Figure 1B illustrates how the AFOs initially improved the knee alignment while standing with the walker (age 76 months). In the pre-surgery phase (Figure 1C, age 88 months), the orthoses failed to control the knee alignment and the internal rotation of the foot progression angle due to progression of the equinus and varo-supination. Figure 1D shows the feet radiography acquired before surgery and intraoperative pictures relative to the tibialis anterior tendon transfer (age 91 months). Figure 1E illustrates the improvement in foot and knee alignment 4 months after surgery (age 96 months), both barefoot and with the orthoses. Additionally, 9 months after surgery (Figure 1F, age 100 months), a relapse of the internal rotation of the foot progression angle with orthoses and knee hyperextension was observed. Nonetheless, the patient was still able to actively resolve varo-supination in sitting position, as shown in Figure 1F. Figure 1G,H show the foot and knee alignment while standing before and 1 month after botulinum toxin injection, respectively, with partial improvement without orthoses. A total of 4 months after botulinum (Figure 1I, age 109 months), no significant change could be noted while standing barefoot, but the malalignment was still flexible and improved with the orthoses. Figure 1J represents the radiographical image of the left tibial fracture (age 111 months). Figure 1K shows the functional situation at the last assessment (age 121 months), with pronated feet while standing barefoot but more positive alignment while standing and no internal rotation while walking with the orthoses.

During the follow-up (FU) period, the patient underwent assessments, whose results are shown in Figure 2: Gross Motor Function Measure-88 (GMFM-88) (Figure 2A), nerve conduction study (NCS) including nerve conduction velocity (Figure 2B), body mass index (BMI) [14] (Figure 2C), clinical evaluation (CE) including the Modified Tardieu Scale (MTS) and passive range of motion (pROM) (Figure 2D) and muscle strength examination according to modified Medical Research Council (mMRC) scale (Figure 2E).

MTS was limited to plantar flexor muscles, with longitudinal assessment of slow pROM at knee flexed (KF) and extended (KE), while the level of spasticity and corresponding fast range of motion (fROM) were measured only before and after botulinum toxin-A (BTX-A) injection. Gait analysis (3DGA) was performed before and after surgery (Figure 3A–D).

Data collected at the Rehabilitation Unit were merged with data collected at the San Raffaele Hospital prior to June 2017 and at subsequent visits. In particular, body mass index (BMI) and nerve conduction study (NCS) is reported. According to the clinical trial protocol [12], only one side was tested for NCS, given that both sides are equally affected by hereditary peripheral neuropathies, and brain magnetic resonance imaging (MRI) was regularly acquired.

### 2.1. First Access to Rehabilitation Unit

In June 2017, the patient was 6 years old (75 months). After the follow-up visit +5 years post-gene therapy, she was referred to rehabilitation because of unrestrained equinus and varo-supination in swing phase and low compliance to her solid ankle-foot orthoses (AFO). The child had achieved the ability to walk independently at 12 months, but at 28 months (1.5 years post-gene therapy) started to show a slowly progressive attitude in knee hyperextension, hyperlordosis, and plantarflexion, which increased with gait velocity. At 4 years and 10 months, she started using AFOs because of the equinus worsening with feet varo-supination.

At the time of her first assessment at the Rehabilitation Unit, the patient could not walk without AFOs. She started leaning on furniture or the walls at home or in the classroom; for longer distances, in crowded environments or on uneven surfaces, she depended on the support of a caregiver. She was deteriorating from Gross Motor Function Classification in MLD (GMFC-MLD) level 1 to level 2 [15]. The gait pattern was characterized by constant hyperlordosis and pelvic anteversion, trunk bending ipsilateral to stance phase, internal rotation of the hips and foot progression angle (FPA), equinus, and varo-supination, which partially resolved in single stance phase and by extensor hallucis longus hyperactivity, knee hyperextension (KH) in mid stance, and stiff knee in initial swing phase. While seated, the patient was able to actively resolve varo-supination. No significant limitation to passive range of motion (pROM) was noticed (Figure 2D). Spasticity was observed at plantar flexor muscles, as level 2 at MTS. Strength was evaluated according to mMRC. It was gluteus maximus 3+, gluteus medius 3+, hamstrings 3+, quadriceps 4, plantar flexors 4, dorsiflexors 3, and adductors 3+ (Figure 2E). Focal laxity was noticed, reaching 20° of hyperextension in both knees. Femoral intrarotation malalignment contributed significantly to FPA, with passive ROM in internal rotation of both hips at 90° and Ryders test 45°. No scoliosis was observed on radiography. Brain MRI showed stable minimal bilateral white matter abnormalities in peritrigonal region and centrum semiovale. Nerve conduction study (NCS) confirmed demyelinating sensory-motor neuropathy, unmodified compared to previous exam 1 year earlier (Figure 2B). Cognitive and language functions were preserved. BMI was between the 25th and the 50th percentile [14] (Figure 2C).

### 2.2. Orthoses (Age 76 Months)

The initial approach was to introduce flexible carbon spring AFOs, customized on cast, with internal adjustments to correct the varo-supination, spring ankle angle 87°, and external calcaneal wedge to counteract recurvatum knee. It appeared lighter and better tolerated than the previous solid AFOs. Moreover, a posterior walker was introduced to achieve independent mobility in the community. Immediate gait improvement was observed: FPA internal rotation and KH were reduced, and the patient regained independence in community walking, no longer requiring caregiver support.

### 2.3. Age 88 Months

The patient came back complaining about falls and again had reduced compliance with AFOs. Clinical evaluation revealed a worsening of adduction and varo-supination of both feet, still passively amendable, with lateral overload, particularly at the base of the fifth metatarsal bone. As this was no longer manageable with AFOs, surgery became necessary to maintain ambulation. GMFM-88 for dimensions D and E were performed, scoring 46% and 14%, respectively (Figure 2A). A gait analysis (Figure 3A–D) was performed, and a multidisciplinary discussion was held to ponder the choice between exclusively soft tissue or bone surgical options. Considering that the deformity (forefoot adduction, supination, and calcaneal varus) was still almost completely passively reducible (dorsiflexion at KF 10° and at KE 0°) (Figure 2D) and that the priority was to minimize immobilization and unloading time to reduce the risk of non-use muscular atrophy, soft tissue functional surgery was preferred.

### 2.4. Surgery (Age 91 Months)

Bilateral surgery was performed, consisting of posterior tibialis lengthening, abductor hallucis, plantar aponeurosis and flexor hallucis longus tenotomy, recession of the extensor hallucis longus to the proximal phalanx of the 1st digit, and tibialis anterior tendon transfer (TATT) to the cuboid, securing it according to the pull-out fixation technique [16,17]. The aim was to solve the forefoot muscle imbalance supporting adduction and supination, together with release of posterior tibialis to counteract calcaneus varus. Ankle-foot plaster casts were positioned for a 2-week period, compatible with standing and walking. Intensive physiotherapy was performed to maintain proximal muscle activity and to immediately restart assisted standing. After the casts were removed, flexible carbon spring AFOs were reintroduced, and the patient continued attending physiotherapy 3 times per week. A daily home program of active exercises and the use of a standing device were also recommended. GMFM-88, limited to dimensions D and E, and a computed 3D gait analysis (3DGA) were performed before and after surgery (Figure 3A–D).

### 2.5. Age 96 Months

Four months after surgery, dorsiflexion pROM had improved (Figure 2D), varo-supination could still actively be resolved while sitting, significantly reduced while standing barefoot, and was well controlled inside the AFOs, but no evident functional gain was detectable at 3DGA (see dedicated section below and Figure 3A–D).

### 2.6. Age 100 Months

Nine months after surgery, a significant improvement was recorded by means of gait analysis (Figure 3A–D) and GMFM (Figure 2A). The gait pattern was still characterized by hyperlordosis and pelvis anteversion, trunk bending ipsilateral to stance phase. Nonetheless, the equinus resolved in single stance phase, and internal rotation of the hips and the FPA were reduced. In the same period, the patient underwent brain MRI and NCS (Figure 2B) as part of the FU protocol at the S. Raffaele Hospital. The MRI was stable; NCS showed a stable nerve conduction velocity (Figure 2B) but a reduction in compound motor action potential (cMAP) amplitude of the right deep peroneal nerve. This could be explained as a consequence of a partial lesion of the distal branch of the deep peroneal nerve, innervating extensor brevis digitorum muscle. This event may have occurred during surgery as a possible neurovascular complication of TATT. Nonetheless, no functional negative relapse was observed, and the foot progression angle remained aligned, both at 3DGA and at observational gait analysis. The BMI exceeded the 50th percentile (Figure 2C).

### 2.7. Age 104 Months

Almost 1 year after surgery, recurrence of varo-supination of the left foot was observed. Persistent internal rotation of the FPA on the left and equinus, which resolved in single stance phase, was observed. Knee hyperextension and stiff knee worsened. Varo-supination on the right side had resolved. Muscle strength was stable (Figure 2E). At clinical exam, the slow pROM at dorsiflexion reduced (10° at KF and 5° at KE on the left, 10° at KE on the right) (Figure 2D), and bilateral distal spasticity had worsened as MTS 2 at KF (−10° left, −5° right) and MTS 3 at KE (−20° left, −10° right). Therefore, a spastic treatment on soleus and left tibialis posterior muscles were considered. To test the consequences of this approach, a diagnostic nerve block was first attempted, injecting 1 cc of lidocaine 2% into the neural branch of the soleus under ultrasound and electrical stimulation guidance. A positive effect was observed in terms of a reduction in spasticity without excessive weakness; bilateral KH resolved, and varo-supination of the left foot decreased. This led us to plan botulinum toxin injections in tibialis posterior and soleus muscles.

### 2.8. Botulinum Toxin-A Injection (Age 104 Months)

Botulinum toxin-A (BTX-A) was injected (Botox 100 U/1 mL) under ultrasound and electrostimulation guidance into the soleus (50 units on the left and 30 units on the right side) and the left tibialis posterior (30 units).

### 2.9. Age 105 Months

A month after BTX-A, passive dorsiflexion improved on the left side (20° at KF, 15° at KE), while no change was observed on the right side (Figure 2D). The spasticity was reduced bilaterally with MTS 0 at KF and 2 at KE (catch at −15° on the left and −10° on the right). A reduction in varo-supination in the left stance phase was observed, but the parents reported increased weakness; they described more difficulty in raising the legs to climb stairs and rolling from a supine to a prone position. At clinical evaluation, a bilateral strength reduction, particularly concerning adductors and gluteus maximus, was confirmed (Figure 2E), as was a slight worsening at GMFM (Figure 2A) and 3DGA (Figure 3A–D); other clinical features were stable, except for a further increase in BMI (Figure 2C).

### 2.10. Age 108 Months

Four months after BTX-A, a clinical examination (Figure 2D,E), GMFM (Figure 2A), and another 3DGA (Figure 3A–D) were performed. The limitation in the left passive dorsiflexion recurred to 10° at KF and 0° at KE. No change in pROM was detected on the right. Strength partially recovered, but the left foot continued to be supinated during standing and walking.

### 2.11. Age 111 Months

MRI and NCS were repeated. The MRI was stable, while the NCS showed a further reduction in cMAP amplitude of the right and left tibial nerve, with concomitant stability of the nerve conduction velocity (Figure 2B) and the other parameters examined at the lower and upper limbs. This result might explain the improvement in right FPA that was later observed. BMI continued to increase (Figure 2C).

### 2.12. Left Tibial Fracture (Age 112 Months)

Just after the follow-up visit at age 111 months, the patient had an accidental distal fracture of the left tibia, which required immobilization with interdiction to load bearing on the left leg for 2 months because of slow recovery from osteopenia. During these 2 months, daily exercises were performed to maintain muscular trophism and ROM (physiotherapy and home program, including exercises in warm water). After 2 months, the use of a standing device and assisted walking was resumed with the carbon spring AFOs and a posterior walker.

### 2.13. Age 114 Months

At the clinical examination 2 months after the fracture, the varo-supination and gait pattern appeared improved, with reduced internal rotation of left FPA, parallel to bilateral plantarflexor strength reduction, while the other findings were stable (Figure 2D,E). The reduction in cMAP amplitude of the tibial nerve observed at the previous assessment at 111 months may have been related to the plantarflexor strength reduction.

### 2.14. Age 121 Months

The gait pattern was stable, but the feet were no longer varo-supinated; conversely, there was valgopronation in both while standing. The pROM at dorsiflexion (Figure 2D) was bilaterally limited to 0° at KE and 10° at KF; an initial contracture of hip flexors of 5° was noted with the Thomas Test. A slight reduction in strength was seen at the level of psoas, quadriceps, and adductors (Figure 2E). No scoliosis was observed. A further increase in BMI was reported, reaching the overweight range (Figure 2C).

### 2.15. Computed Gait Analysis Findings

Three-dimensional gait analysis was performed by means of a Vicon^®^ system (Oxford Metrics Group, UK). The system was equipped with 8 optoelectronic cameras, 2 force plates (AMTI, Watertown, MA, USA), and 2 video cameras. A 10 m walkway allowed the patient to reach and maintain a constant self-selected walking speed during acquisitions. The marker set followed the Total3DGait protocol. Kinematic data were recorded with the patient’s posterior walker. The patient wore her ankle-foot orthoses and shoes. Walking speed (self-selected, normalized to height) and stride length (normalized to height) were taken into consideration. We analyzed sagittal plane kinematics of the knee and foot progression angle (FPA). The Gait Profile Score (GPS) and the Gait Variable Score (GVS) were calculated. The GPS represents the root mean square difference between a particular gait trial and averaged data from individuals without a gait impairment [18]. It was developed to summarize data on kinematics and to report them with a synthetic score for the overall gait pattern or separated for the right and the left side. The GPS is based on 9 kinematic variables. It can be broken down to provide the GVS, which describes the magnitude of the deviation of those variables across the gait cycle [19].

Spatial–temporal parameters (STP) indicated a trend of general improvement (Figure 3A) from 88 to 121 months, both in speed (from 35 to 52%h/s) and in stride length (from 66 to 72%h). Best performance was reached at 9 months after surgery.

In addition, global GPS index (Figure 3B) showed a decrease (from 19.9 at age 88 months to 15.2 at 121 months), and therefore a significant improvement, considering that the minimal clinically important difference has been calculated as 1.6 [19]. However, trends were different when the left and right sides were considered separately. Right GPS revealed a progressive improvement in kinematics, from 21.1 to 14.6. On the left side, GPS worsened up to 109 months (from 16.3 to 21.1), then suddenly improved at 121 months, reaching 14.6. This difference was mostly due to the modification of FPA (Figure 3C,D), which decreased from 48.6 to 10.4 for the right foot (a clear reduction in internal rotation, mainly during stance phase) and increased from 24.8 to 48 (increase in internal rotation), with a final decrease on the left foot to 19.2 at the last assessment. Additionally, the Knee Flexion item of GVS significantly improved, from 28.6 to 13.1 on the left and 25.1 to 20.4 on the right (from age 88 to 105 months). In both cases, it was possible to identify a reduction in maximum hyperextension in single stance (from 18° to 10° on the left and from 18° to 5° on the right) and a more effective flexion in swing (the peak increased from 48° to 58° on the left and from 38° to 58° on the right). Most significant right and left kinematics are presented in Appendix A, respectively.

## 3. Discussion

HSPC-GT Untreated MLD subjects similar in age and subtype to the examined patient never learn to walk independently or lose their walking ability by that age [3,8]. Thanks to HSPC-GT, most patients show normalization of motor development and stabilization of motor function according to GMFM or a delay in the rate of progression of motor dysfunction [13]. The patient had a prolonged life expectancy with a reduction in disease progression. HSPC-GT significantly improved the prognosis, introducing the challenge of managing and preserving the residual functions. In fact, 5 years post-HSPC-GT, the patient complained about increasing gait difficulties caused by a residual disease burden associated with secondary deformities and peripheral demyelinating neuropathy overlapping the central pyramidal tract involvement. An integrated orthopedic and rehabilitative approach, therefore, became necessary; this approach proved to be effective in preserving ambulation.

Scoliosis, hamstring contractures, and equinus are described as the most common orthopedic manifestations in leukodystrophies [6]. Our patient presented neither scoliosis nor hamstring contracture. Bilateral equinus, initially recorded at gait analysis, was more evident on the left side and during the stance phase, accompanied by hyperextension of the knees and stiff knee gait. Nonetheless, no significant equinus contracture was observed, with dorsiflexion passive ROM always reaching at least the neutral position when the knee extended. The most disabling initial defect was bilateral varo-supination, which led to discomfort because of internal rotation of the foot progression angle and intolerance to AFOs, which were essential to reduce recurvatum of the knees, to ensure stability in the stance phase, and to avoid equinus in the swing. Orthopedic surgery, the last option we considered, became necessary to prevent the loss of walking ability. The orthopedic and rehabilitative approaches in MLD reported in the literature are only anecdotal, particularly because of the speed of progression of the early onset MLD variants when untreated, making any intervention ineffective in the majority of cases. Other authors have recommended to timely approach the deformity in MLD children treated with gene therapy [6]. Functional neuro-orthopedic surgery that takes into account the muscle imbalance is advisable, as it is the primary cause of malalignment recurrence. The transfer technique was chosen for our patient to resolve muscle imbalance and to permit immediate load and recovery of walking. Surgery initially resolved varo-supination on both sides and restored compliance to AFOs. Gait speed, stride length, and GPS improved, in particular, because of the bilateral improvement in the knee and ankle kinematics; the stiff knee and the hyperextension of the knee decreased bilaterally. The varo-supination was resolved on both sides at cast removal, but by the time of the gait analysis (4 months after surgery), the internal foot progression angle had improved only on the right side, while the varo-supination recurred on the left side, despite substantial stability of NCS and MRI findings. Both feet could easily be aligned by passive mobilization, and passive dorsiflexion ROM improved. The recurrent left varo-supination with internal FPA may be explained by the recurrence of muscle imbalance with tibialis posterior overactivity and increased spasticity of left plantar flexors, exasperated by a growth spurt. A worsening trend on the left side resumed, while the right-side kinematics slowly improved. This led us to consider botulinum toxin injection, as suggested by several reports to treat spasticity in MLD subjects [20,21,22]. No negative outcome was observed after the diagnostic anesthetic neural block. Nonetheless, a general weakness was reported by the parents after botulinum toxin injection, confirmed by a muscle strength decrease on the mMRC scale, reduced speed, stride length, and GPS, which recovered at the following assessment 5 months after the injection. Relapsing manifestations of MLD may have contributed. Nonetheless, the trend of the functional worsening over time suggested that the botulinum itself may have played a determinantal role. This may have been due to the diffusion of BTX-A to surrounding muscles, despite reduced dilution (100 U/1mL) and the use of ultrasound and electrical stimulation to correctly localize the targeted muscles. The recovery time from collateral weakness lessened the benefit obtained in terms of reducing equinus, varo-supination, and internal FPA. Similar findings, with no overall functional improvement, emerged in studies on subjects with hereditary spastic paraplegia. Those authors reported secondary muscle weakness [23,24,25], which, in most cases, outweighed the advantages related to spasticity reduction [23]. This suggests caution in using botulinum toxin to treat spasticity in similar cases, although several authors have reported it as a possible approach in MLD subjects [20,21,22]. Equinus at gait analysis was bilaterally reduced after botulinum toxin injection. The dorsiflexion pROM also improved for a short time, but it soon worsened bilaterally, reaching no more than 0° at KE. Finally, the internal FPA on the left side resolved after the left tibial fracture. This gain on the left foot alignment after plastering and after an almost 2-month ban on weight-bearing subsequent to the tibial fracture confirmed the role of muscle imbalance.

During the immobilization periods after surgery and after the fracture, an effort was made to maintain overall muscular trophism and strength with regular physical exercise at home and with the physiotherapist. Nonetheless, a trend toward reduction in muscle strength was recorded at the following visits. In parallel, a significant increase in BMI was recorded during FU. Strength reduction and being overweight may have been partially due to reduced physical activity during the COVID-19 emergency [26], as observed in the overall population [27]. Overweight and laxity are known risk factors for flatfoot deformity, particularly in children with delayed motor development [28,29,30]. Therefore, it is plausible that the increased BMI in the patient may have induced foot overload and contributed, together with laxity, to structural changes in valgopronation.

Furthermore, being overweight may have a role in inducing relative muscle weakness. In fact, it is known that obese individuals have reduced maximum muscle strength relative to body mass in their anti-gravity muscles compared to non-obese persons [31]. Moreover, reduced muscle strength, normalized to body mass, has been found in young survivors with overweight after hematopoietic cell transplantation [32]. Being overweight was also associated with lower physical activity levels than those of peers with normal nutritional status [32].

The bilateral reduction in cMAP amplitude of the tibial nerve at the last follow-up may have been due to the local modifications of the foot (valgopronation posture) and to the altered plantar load during gait. This hypothesis is supported by the stability of NCS upper limb parameters and lower left peroneal nerve parameters during the last exam at 121 months. Nonetheless, a reduction in the cMAP amplitude has been described in untreated MLD [33], and axonal loss and a reduction in the number of myelinated fibers have been reported among peripheral nerve abnormalities [34]. Therefore, a slight and very distal progression of the already-known demyelinating neuropathy, which appears to be the most refractory to therapy, cannot be totally excluded [34]. Further studies are needed to understand peripheral nerve alterations in MLD subjects treated with gene therapy.

The major clinical outcome was a progression from varo-supination to valgopronation first of the right foot, then of the left foot as well. This may be attributed to the combination of the reduced supinating moment due to surgical correction of the muscle imbalance and BMI increase with underlying feet laxity. However, prior to surgery, varo-supination led to foot ulcers and loss of ambulation, necessitating surgical intervention. Conversely, valgopronation was comfortably managed by means of orthoses. The overall GPS, gait speed, and stride length improved over the course of FU, despite intermediate worsening phases before and after botulinum toxin injection, respectively, due to increased equinus and general weakness. This confirms the effectiveness of a combined, individually tailored approach in children showing a residual disease burden after undergoing disease-modifying advanced experimental treatments.

## 4. Conclusions

The take-away lessons of this case report are:—Muscle imbalance and increased BMI have a determinantal role in gait deterioration in the growing phase of MLD patients treated with HSPC-GT and experiencing a stabilization/slowing of disease progression;—Caution is recommended when considering botulinum toxin injection in early-onset MLD subjects suffering from both CNS and PNS involvement because of the risk of inducing overall weakness as a collateral effect, which outweighs the benefits of spasticity reduction;—A combined approach including conservative rehabilitative options, such as orthoses, walking devices, exercises, and functional orthopedic surgery, is advisable in order to maintain gait ability;—Based on the demonstration that most patients treated presymptomatically with HSPC-GT show a stabilization or delay in the rate of progression of motor dysfunction and maintain long-term walking capability, strict monitoring of gait disturbances through a multidisciplinary team approach and multiple instrumental parameters (clinical exam, gait analysis, NCS, BMI, etc.) is mandatory to maintain long-term gait capability, prevent deformities and pain, and guarantee independence in daily life;—Further studies are needed to understand peripheral nerve alterations in MLD subjects treated with gene therapy and its impact on clinical outcomes.

## Figures and Tables

**Figure 1 jpm-13-00637-f001:**
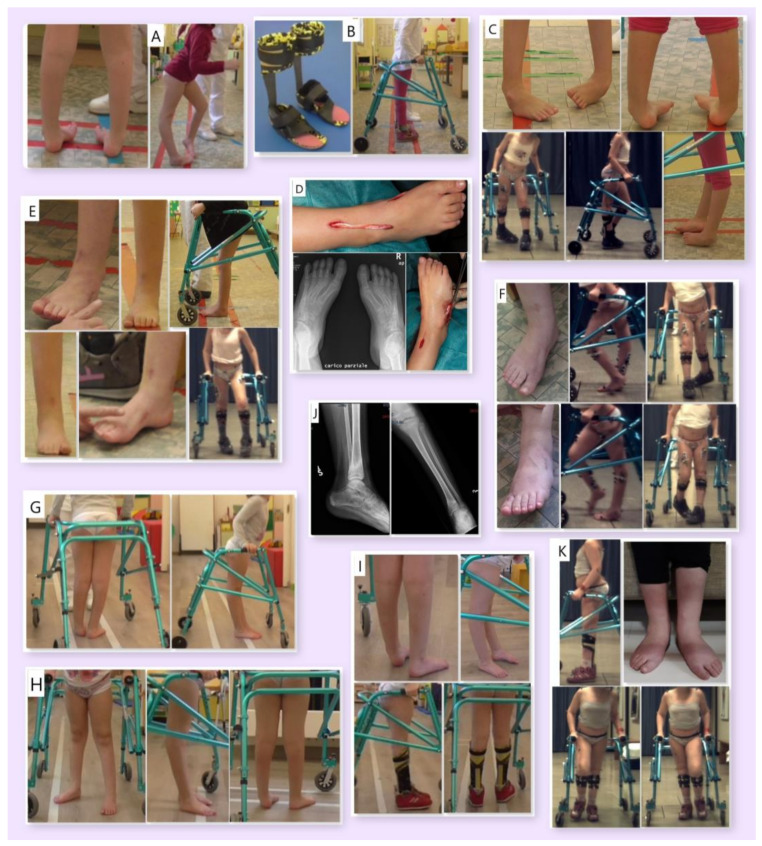
Images of the patient: at first access to the rehabilitation unit ((**A**), age 75 months); AFOs and walker ((**B**), age 76 months); pre-surgery ((**C**), age 88 months); intraoperative ((**D**), age 91 months); 4 and 9 months after surgery, respectively ((**E**), age 96 months; (**F**), age 100 months); pre-botulinum toxin injection ((**G**), age 104 months); 1 month after botulinum toxin injection ((**H**), age 105 months); 4 months after botulinum toxin injection ((**I**), age 109 months); left tibial fracture ((**J**), age 111 months); last assessment ((**K**), age 121 months).

**Figure 2 jpm-13-00637-f002:**
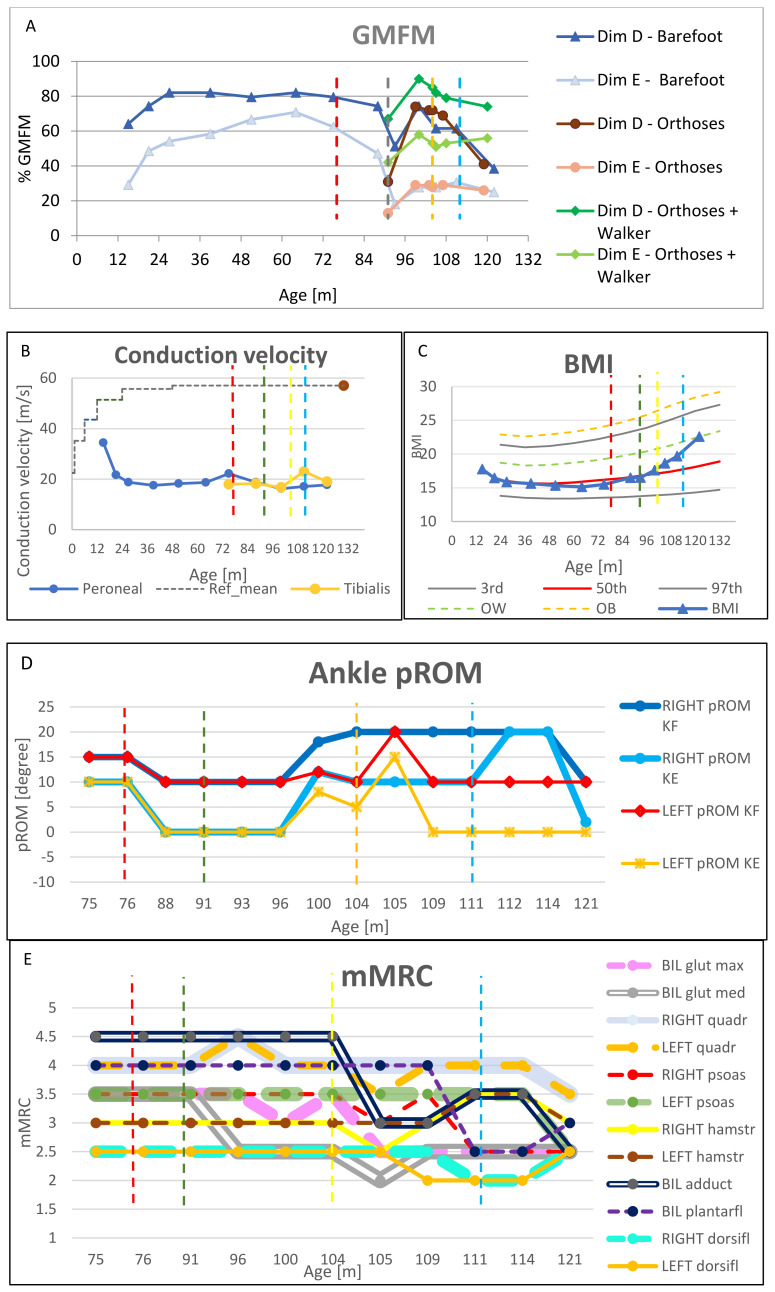
Clinical and instrumental assessments performed over the follow-up period, at different months of age. Time at which interventions and fracture occurred is represented with vertical dot lines: respectively, AFO in red, surgery in green, botulinum in yellow, and tibial fracture in light blue. (**A**): Gross Motor Function-88 (GMFM-88) dimensions D and E, in different conditions: barefoot, with orthoses, with orthoses and walker. (**B**): Conduction velocity relative to the right leg. (**C**): BMI of the patient, with Italian growth norms for females [14], related to the 3rd, 50th, and 97th percentile and extra-centiles for overweight (OW) and obesity (OB). (**D**): Lower limb passive range of motion (pROM), according to Modified Tardieu Scale: slow pROM to assess muscle stiffness and contractures, fast pROM to assess spastic reaction (which identified the range at which the catch was evoked). (**E**): Lower limb strength assessment by means of modified Medical Research Council (mMRC) scale.

**Figure 3 jpm-13-00637-f003:**
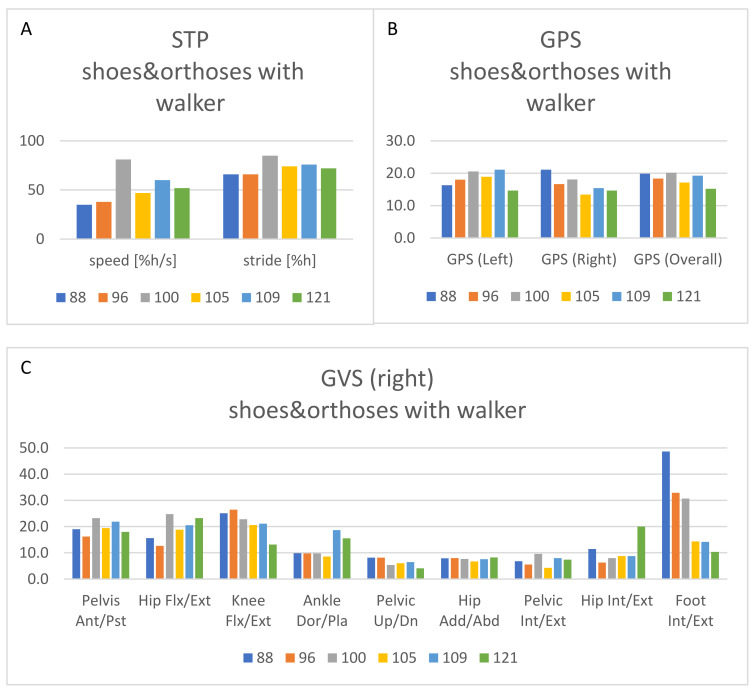
Results from the 3DGA over the follow-up period at 88, 96, 100, 105, 109, and 121 months of age. (**A**): Spatiotemporal parameters (speed and stride length, both normalized to the patient’s height), referring to gait analyses acquired with orthoses and walker. (**B**): Gait profile score (GPS) for the left and right side and overall, referring to gait analyses acquired with orthoses and walker. (**C**): Gait variable score (GVS) for the right side, referring to gait analyses acquired with orthoses and walker. (**D**): Gait variable score (GVS) for the left side, referring to gait analyses acquired with orthoses and walker.

**Table 1 jpm-13-00637-t001:** Timeline of assessments and interventions.

Age (Months)	Interventions	Assessments *				
15	HSPC infusion	GMFM-88	NCS			BMI
21		GMFM-88	NCS			BMI
27		GMFM-88	NCS			BMI
39		GMFM-88	NCS			BMI
51		GMFM-88	NCS			BMI
64		GMFM-88	NCS			BMI
75		GMFM-88	NCS		CE	BMI
76	AFO and walker					
88		GMFM-88	NCS	GA	CE	BMI
91	surgery					
93		GMFM-88				BMI
96				GA	CE	
100		GMFM-88	NCS	GA	CE	BMI
104	BTX-A				CE	
105		GMFM-88		GA	CE	BMI
109				GA	CE	
111		GMFM-88	NCS		CE	BMI
112	left tibia fracture					
114					CE	
121		GMFM-88	NCS	GA	CE	BMI

Legend: AFO: Ankle Foot Orthosis; BTX-A: Botulinum Toxin-A; GMFM-88: Gross Motor Function Measure-88; NCS: Nerve Conduction Study; 3DGA: Three-Dimensional Gait Analysis; CE: Clinical Evaluation; BMI: Body Mass Index. * The slight difference in GMFM and 3DGA schedules is mostly due to difficulties matching family availability with access to the hospital.

## Data Availability

The data presented in this study are available on request from the corresponding author. The data are not publicly available due to ethical reasons.

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
