# Peer review of "Preserving Ambulation in a Gene Therapy-Treated Girl Affected by Metachromatic Leukodystrophy: A Case Report"

_jpm, 2023, doi:10.3390/jpm13040637_

Round 1

Reviewer 1 Report

This paper illustrates an interesting observation of clinical course of post-gene therapy MLD management. 

However, to facilitate readers' easy understanding, it would better to revise the manuscript to rectify quite a few misnomers, discrepancies between diagrams and labelings and frequent errors in language expression.

The details are as follows:   

Table 2: mixed use of NCS and NCV, may need to unify and explain the full terminology in the caption 

Fig 2E: difficult to read, some graph (for example, purple line) was not linked with any muscle

Fig 3: caption is not easy to understand

line 156: "first signs of limb spasticity" what do the authors mean?  

line 157: "feet intrarotation" may not be a proper medical terminology.  

line 238: What do you mean "exam slow pROM"?

line 244: What do you mean "positive relapse was observed"?

line 258-259 Gluteus maximum may not be a proper terminology. 

line 226, 269 Compound motor action potential may not be a proper terminology. 

line 284, 285: The authors described ongoing bilateral ankle plantar weakness. Would better link with NCS findings mentioned above.  

line 287-288: the changes in gait pattern and alignment of feet might have been scrutinized from the other perspectives, i.e. natural development from infancy to childhood, such as genu varus and subsequent genu valgus.    

Line 325-327: difficult to understand the meaning

line 367-370: Given the small dose of botox total 110 unit injected to bilateral soleus and left posterior tibialis under the dual monitoring of ultrasound and electrical stimulation, the author's interpretation that the general weakness is exclusively a sequel of botox injection might be insufficient. Rather the possibility of relapsing or fluctuating manifestations of MLD could be added.      

line 384-385 The author may provide the reference to support the statement that increased BMI can entail valgopronation. 

line 409-412 difficult to understand. The author's take home message highlighting that BMI 23, borderline or very mild overweight, could be one of the principal causes of gait deterioration of MLD patient is questionable. The author should attach other references to bolster this conclusion.    

The author may need to maintain consistency in writing, whether to use British or American English. That said, professional proofreading may help to deliver the message effectively.  

Thanks. 

Reviewer 2 Report

In general, the case report looks coherent with sufficient details. I have the following comments:

1.     I think the authors can improve the background and include some recent advances in gene therapy methods.

2.     Authors should discuss Figure 1 A in a bit more detail than just mentioning “Clinical and X-ray images of the subject are represented in figure 1”. I think it's an important figure and deserves a paragraph at least.

3. It would be more readable if the figure panel names such as 2A, 2B, etc are placed on the top left instead of the bottom left of each panel. Currently, it seems to be distracting and merging with the axis labels

Reviewer 3 Report

This is an interesting case of a patient treated with HSPC-GT in which essential observations and recommendations are given. Therefore, this manuscript may add relevant information with regard to Metachromatic leukodystrophy disease progression. 

Nevertheless, the manuscript is wordy, and using capitals is rare and can be corrected. The images are difficult to understand and the titles are not well separated. Could authors improve pictures and their explanations? 
